# A Small Area Estimation Method for Investigating the Relationship between Public Perception of Drug Problems with Neighborhood Prognostics: Trends in London between 2012 and 2019

**DOI:** 10.3390/ijerph18179016

**Published:** 2021-08-26

**Authors:** Arun Sondhi, Alessandro Leidi, Emily Gilbert

**Affiliations:** 1Therapeutic Solutions (Addictions), London W1K 1QW, UK; 2Statistical Services Centre Ltd., Reading RG30 2TL, UK; sandroleidi1960@gmail.com; 3Evidence and Insight, London Mayor’s Office for Policing and Crime, London SE1 2AA, UK; Emily.Gilbert@mopac.london.gov.uk

**Keywords:** Small Area Estimation, drugs, public attitudes

## Abstract

The correlation of the public’s perception of drug problems with neighborhood characteristics has rarely been studied. The aim of this study was to investigate factors that correlate with public perceptions in London boroughs using the Mayor’s Office for Policing and Crime (MOPAC) Public Attitude Survey between 2012 and 2019. A subject-specific random effect deploying a Generalized Linear Mixed Model (GLMM) using an Adaptive Gaussian Quadrature method with 10 integration points was applied. To obtain time trends across inner and outer London areas, the GLMM was fitted using a Restricted Marginal Pseudo Likelihood method. The perception of drug problems increased with statistical significance in 17 out of 32 London boroughs between 2012 and 2019. These boroughs were geographically clustered across the north of London. Levels of deprivation, as measured by the English Index of Multiple Deprivation, as well as the percentage of local population who were non-UK-born and recorded vehicle crime rates were shown to be positively associated with the public’s perception of drug problems. Conversely, recorded burglary rate was negatively associated with the public’s perception of drug problems in their area. The public are influenced in their perception of drug problems by neighborhood factors including deprivation and visible manifestations of antisocial behavior.

## 1. Introduction

Measuring how the general public perceives the extent and nature of criminal and anti-social behavior has been used to inform policymakers on whether crime is seen to be increasing [1] and on attitudes towards the police [2,3]. Public perspectives towards antisocial and criminal behavior have also been viewed as a possible prognostic to poor physical [4] and mental health [5,6,7,8]. The use of illicit substances has also been placed within an antisocial behavioral framework such that drug use has been shown to be strongly associated with knife crime [9]; assault, [10] including sexual assault [11] and domestic abuse [12]; other forms of violence, including gang-related behaviors [13]; and acquisitive crimes, encompassing burglary and theft [10,14,15].

Neighborhood characteristics have been identified that attempt to explain differential rates of crime through the extent of social disorganization within an area [16] and how community disorder such as vandalism and open drug use can be perceived as physical manifestations that provide ‘signals’ of localized antisocial behavior [17,18,19]. Public perceptions of crime and antisocial behavior have also been viewed as subject to a ‘perceptions gap’, whereby what the public perceives as the extent of crime is not in line with actual reported levels [20,21] but rather reflects generalized social anxiety.

The role of illicit drug misuse as a contributory factor underpinning local crime rates within a wider social exclusion framework has been articulated [22,23,24]. In addition to antisocial behavior, public perceptions may be closely aligned with wider public health concerns whereby visible manifestations of illicit drug misusing behavior have been shown to be linked with injecting drug use and the use of social space within communal housing areas [25,26]. Moreover, public health concerns can also be shown to coexist with overt drug-dealing that would exacerbate illicit drug use, leading to a conflagration of health-related issues including initiation, risky behaviors such as injecting or sharing of needles and overdose. In addition, within a community, visible signs of illicit drug misuse often coalesce around drug markets which exist as geographic locations in which sellers engage with buyers in ‘open’ settings within public spaces with links to public transport or in ‘closed’ settings or private dwellings [27]. The site for drug markets has been shown to be closely aligned to areas of social and economic deprivation [28,29]. Within the United Kingdom (UK), urban drug markets are synonymous within highly visible ‘user-dealers’ who are well-established within a local community [30]. Recent research has argued that the drug markets within London and throughout the UK are highly competitive with a commensurate level of associated gang-related violence [31,32].

Given the importance of drug use as a precursor for criminality and antisocial behavior, little is known about the correlates of the public’s perception of local drug-related problems. One study in New Zealand found fear of crime strongly associated with drug-related offences which were exacerbated by the extent of how the area was considered fragmented [33]. The aim of this study was to examine time-varying factors that are associated with changing annual perspectives on the extent to which drug use is a ‘problem’ within the Greater London area from 2012 to 2019. It is hypothesized from multidisciplinary prognostics that encompass official police recorded crime rates that neighborhood factors linked with social disorganization are associated with the public’s perceptions of drug problems within their local area. The analysis will also examine whether any drug-specific issues and interventions such as the extent of engagement with local drug treatment, the rate to which people are hospitalized as an acute in-patient for drug misuse and the level of drug poisonings are mediating factors in the public’s perception.

## 2. Materials and Methods

Data on public perceptions of drug use were obtained from the Mayor’s Office for Policing and Crime Public Attitude Survey (PAS), which is a survey of randomly-selected London residents aged 16 or over. The PAS obtains a representative sample of 12,800 interviewees annually in each of London’s 32 boroughs (excluding the City of London) per year. For this study, the binary dependent variable extracted from PAS was a question asking: ‘how much of a problem are people using or dealing drugs?’, with the respondent asked to think about their local area when answering the question. The question specifies the local area to be within 15 minutes’ walk from the respondent’s home residence. Respondents are given the option to respond with ‘Very big problem’, ‘Fairly big problem’, ‘Not a very big problem’ and ‘Not a problem at all’. For the purpose of this analysis, the variable was coded dichotomously, with ‘very big problem’ and ‘fairly big problem’ coded to indicate the respondent thinks drugs are a problem in their local area, and the second two coded to indicate the respondent thinking drugs are not a problem in their area.

The explanatory variables were derived from open sources and apportioned into three domains. The first examined social correlates derived from the literature relating to the local area, such as age structure [34], focusing on the percentage aged 15 to 29 years; ethnicity [35], including the percentages who were defined as non-white and non-UK-born; mid-year population estimates; and the overall level of deprivation as indicated by the English Index of Multiple Deprivation (EIMD). The second domain focused on levels of recorded crime for burglary, arson and criminal damage, drug offences, robbery, sexual offences, theft, vehicle offences, violence and weapons offences [36]. The third domain included drug-specific prognostics that focused on acute drug-related needs within an area including the extent of specialist treatment demand (the number of people within an area reported as receiving drug treatment), the number of drug poisonings and the number of acute in-patient hospital admissions related to drug misuse. All data were accessed from publicly available open sources (https://data.london.gov.uk accessed on 23–29 July 2020). The datasets were merged by borough and year, creating a final dataset of 256 records pertaining to 32 London boroughs, each contributing 8 consecutive years of PAS data, from 2012 to 2019.

As PAS data require a support distribution appropriate for a binary outcome, a binomial distribution with the total number of respondents was used as the denominator to allow for the modelling of the probability of a respondent replying that drugs were a problem. Borough was defined as a random effect, making this a Generalized Linear Mixed Model, or GLMM [37]. GLMM is a model-based technique suitable for Small Area Estimation [38] and exhibits the shrinkage property, that is, it estimates a limited number of fixed regression parameters, while it is capable of computing borough-specific predictions. The GLMM needs to account for the nature of the records of the repeated measures, as each borough provided a time series of eight consecutive yearly records. To evaluate the time trend within each borough, a random coefficients model was fitted [39] by adding two random effects, an intercept and a slope, which were allowed to be correlated via an unstructured covariance matrix. Following Stroup [37], the GLMM for binary data can be written in matrix notation as:logit(µ) = X(β) + Z(α)
where logit() is the link function for binomial data, μ is the outcome, i.e., the predicted probability of a PAS respondent saying drug-related crime is a problem, and where X is a design matrix of fixed effects, Z is a design matrix of random effects, β is a vector of fixed regression coefficients and α is a vector of random regression coefficients. The vector α is assumed to be normally distributed with expectation zero and covariance matrix G, a block diagonal matrix with 2 by 2 symmetric blocks with 3 covariances. As boroughs were grouped into inner and outer London areas, there are two different symmetric blocks, for a total of 6 covariance parameters. The input to the left-hand side of the above equation is the outcome μ, formed as numerator over denominator from the PAS results of each of the unique combinations of 32 boroughs by 8 survey years, yielding 256 rows of data. The inputs to the right-hand side of the above equation were as follows: the 19 predictors recorded at the borough level for each year of the PAS survey go into the design matrix X, while Z is a design matrix assembled by the statistics package, with 8 rows and 2 columns of non-zero entries for each borough, whose purposes are to predict borough-specific time trends. Outputs from the above equation are all quantities presented in tables and figures in the paper.

The PAS response to local drug perceptions was modeled with 19 predictors fitted as fixed effects, including an integer from 0 to 7 representing the eight consecutive years in the temporal series for estimating the time trend. A dichotomous categorical prognostic was created with two levels across inner London (comprising 12 boroughs) and outer London areas (comprising 20 boroughs). The difference between inner and outer London is a reflection of historical administrative units but over time has come to reflect different socioeconomic compositions. Although inner London has some of the richest boroughs in the world, there exist high concentrations of urban deprivation within the inner ring of London, relative to outer London. The interaction between year and area was used as an effect modifier, allowing the time trend to differ between the inner London and outer London area. EIMD was determined by the weighted mean score. The recorded crime categories were expressed as a rate per 1000 people with weapon offences expressed as a rate per 10,000 due to the infrequency of this offence type. Such rates were computed using the borough mid-year population estimates released annually by the Office for National Statistics. Data for the City of London were excluded due to its function as a financial center compared to other residential areas.

A random intercept and a random slope were included as random effects clustered by borough. These two random coefficients allow for a borough-specific intercept and trend over time and can also be correlated; hence, they have an unstructured covariance pattern with three components. To allow extra flexibility in time trend between boroughs, a heterogeneous covariance matrix was fitted by London area, that is, the three variance components were estimated separately for inner London and outer London, yielding a grouped unstructured pattern with a total of six covariance parameters. A GLMM was fitted using PROC GLIMMIX within SAS OnDemand for Academics.

To obtain the borough-specific random effect and hence to obtain borough-specific predictions, the GLMM described above was fitted using an Adaptive Gaussian Quadrature method with 10 integration points [37]. Then, to estimate the time trend across inner and outer London, averaged over the boroughs in each area, the same GLMM was refitted using a Restricted Marginal Pseudo Likelihood method; this is also known as population-averaged model [40]. Though recent marginalization proposals derive the implied marginal effects without refitting the same GLMM by a different method [41], such an approach was not implemented in this analysis.

## 3. Results

The proportion of survey respondents who felt drugs were a problem was modeled by a GLMM in an aggregated format (numerator/denominator), and the mixed binomial model converged successfully. Predicted percentages of public perception at the borough level were obtained from the conditional model. Predictions were formed by adding the regression coefficients of both fixed effects and random effects pertaining to the prognostic profile of each combination of borough and year stored in the database. The predicted responses are presented as line charts paneled by borough across London (Figure 1). The Odds Ratios quantifying the time trend by borough (Table 1) shows that the estimated time trend in public perception was statistically different from zero in 17 boroughs. The estimated trend increased in 30 out of the 32 London boroughs, as illustrated by the first 30 tabulated boroughs showing an Odds ratio larger than ‘1’. It can be said that in these boroughs, the public perception that drugs were a problem truly increased over the 8 years in question. These boroughs can be shown to be located overwhelmingly in the north of London. The steepest increases were recorded across North London in Harrow (Odds Ratio (OR) 1.33), Brent (OR 1.22), Waltham Forest (OR 1.20) and Barnet (OR 1.19). The shallowest increase was noted in the east of London in Barking and Dagenham (Odds Ratio 1.01). Conversely, the South London boroughs of Merton (OR 0.99) and Bromley (OR 0.98) reported a slight decrease in the trend of people stating that drugs were a problem in their area.

The analysis examined whether the trends identified in the public’s perception of drug problems were affected by location (inner and outer London). The estimated time trend was shown be statistically different from zero across both inner and outer London, such that the trend in increased rates of perceived drug problems holds across inner and outer London (Figure 2 and Table 2), albeit at a slightly higher rate across inner London (OR 1.104) relative to outer London (OR 1.099).

The next phase of the analysis examined the population-averaged fixed effects whereby the three prognostic domains (socio-demographic attributes, recorded crime types and drug specific measures) were modeled to the rate of public perceptions of drug use in each London borough (Table 3 and Table 4). A Type III test for fixed effects was included to measure the significance of each partial effect (Table 4). Four prognostics were shown to be significantly associated with changes in public perception, two being from the socio-demographic attributes of a borough and two from specific recorded crime. No drug issue prognostics (treatment, drug-related poisoning and in-patient hospital admissions) were shown to be significantly correlated with changes in public perception on drug problems in their area. The percentage of the population that was non-white was shown to be a significant factor, such that an increase of 1 percentage point in this demographic correlated with an increase of 1.01 in the odds of public perception of local drug problems; with EIMD, an increase of 1 unit in this score also correlated with an increase of 1.02 in the odds of the public’s perception of local drugs problems. For specific crimes recorded by the police, it was estimated that an increase of 1 in a thousand in burglary rate correlated with a decrease of 0.96 in the odds of public perception of drug problems, and for recorded vehicle crime, an increase of 1 in a thousand in vehicle rate correlated with an increase of 1.018 in the odds of public perception of drug problems.

## 4. Discussion

This study has shown that across London, between 2012 and 2019, there was a statistically significant increasing trend in the percentage of interviewees perceiving drugs to be a problem in their neighborhood and that this trend is consistent across both inner and outer London. It is hypothesized that the public health implications of increased perceived drug problems in an area will include a greater number of health-related issues such as higher rates of treatment demand, hospital in-patient admissions and drug poisonings. Furthermore, we have been able to show significant increases in specific London boroughs, clustering in North London. It is unclear why North London residents show a higher perception of drug problems relative to South London. This disparity in perception, which may reflect other issues such as differential approaches to policing, that are localized into smaller unit areas across London [42]. Despite national trends in England and Wales showing that interviewees who perceive crime as a problem have fallen from 79% in 2012 to 60% in 2016 [1], the perception of drugs as a local problem has risen significantly throughout most of London, and it has been considered elsewhere that these negative perceptions may reflect cognitive processes unique to the London population [43], or it may indicate that perceptions of drug-related problems may have a different cognitive etiology.

The explanatory analytical framework attempted to divide the prognostics into three domains. The first focused on socio-demographic neighborhood measures, and two measures were found to be significantly associated with increased perceptions of drug problems in an area. The percentage of the population that were recorded as non-white and the level of social deprivation in an area were shown to be prognostic of people perceiving drug problems in their neighborhood. This is broadly consistent with the social disorganization theory [16], and the findings support the contention that social deprivation is a function of public perception [35,43,44], although the finding of being non-UK-born as a marker for increased perception of local drug problems is divergent to international research [45]. Links between recorded crime and components of social disorganization including deprivation and ethnicity have been shown to be true for burglary, robbery, vehicle crime and rates of violent crime across London [46,47].

In the second component of the analysis, we attempted to determine whether there was any association between recorded crime types and trends in public perceptions for drug problems. This was an attempt to examine different crime types rather than a single, overall measure of crime. Unexpectedly, recorded drug offences in an area were not associated with public perceptions of drug problems. This may reflect local police activity focusing on lower order cannabis possession as opposed to substances more associated with criminal activity such as Class A drugs. Cannabis use for personal consumption may be less visible to the wider community [48]. Visible cues of local disorder have been shown to influence public perceptions of crime and antisocial behavior [49], specifically for drug-related problems [50,51]. This may also explain why no other drug-related measure was associated with perceived drug problems. The number of people in specialist drug treatment, the number of drug poisonings and the number of acute in-patient hospital admissions were all not associated with perceived levels of drug issues. We may therefore hypothesize that these measures reflect personal health concerns that will be less visible to the wider community.

Our finding of burglary and vehicle crime as the only two offences associated with increased perceptions of drug problems may also reflect the range of visible signals available to local populations. Vehicle offences include aggravated vehicle taking, theft from and of a vehicle and interfering with a vehicle. These offences are likely to be evident to the wider community, and this crime type, incorporating car vandalism, has been noted in public perceptions of drug problems [51,52]. Visible cues including arson and criminal damage were, however, not found to be significantly related to public perceptions of drug problems, suggesting that public visibility can only be a partial explanation. In addition, offences including violence against people and possession of weapons were not found to be associated with public perceptions of drug-related issues, which is also surprising given the relationship between acts of violence and open drug markets [22,23,24]. Further work is required to delve more deeply into how and why the public perceives social issues to be linked with each other. The finding that increased levels of burglary are negatively associated with perceived neighborhood drug problems may be considered counterintuitive given the relationship between substance use and acquisitive crime [15,53,54]. However, in recent years there has been a noted decrease in recorded burglaries largely due to improved security [55], leading to a hypothesized ‘new’ type of burglar [56]. In this context, international research has identified a shift of burglaries away from more deprived areas to wealthier locations due to an increased number of inner-city apartment blocks with improved security arrangements. Therefore, it has been argued this has led to a shift in focus to more amenable housing stock for burglary [57]. It is unclear whether this finding holds true for London, and further work is required to explore whether this pattern can be confirmed.

### Limitations

This is one of the few studies that focus on a specific component of people’s perception of crime and antisocial behavior by focusing on the extent to which people perceive there is a drug problem in their neighborhood. We also have been able to include time-varying prognostics to identify correlates with changing public perceptions. Access to these datasets has allowed for deployment of small area methods [58], in this instance at borough-level in London, to identify areas with variable changes in public perception. This differs from most studies of public perception that rely on single, cross-sectional analyses of national household surveys often aggregated to a higher administrative level.

A number of limitations should be noted. Although public perceptions were included as an outcome variable, there was no additional information on the characteristics of the people surveyed. Further attention should be focused on the interplay between individual and neighborhood characteristics. We were also limited to the official recorded aggregated crime categories which may miss specific types of offending and antisocial behavior that may be relevant to local perceptions of drug problems (such as graffiti). The use of household surveys has been critiqued to suffer in recent years from non-response and other biases that limit their generalizability [59] and in particular their usefulness in quantifying fear of crime [60]. The PAS collates responses of people’s perception to within a 15-min walk of their home and then aggregates their responses to a convenient administrative borough-level, known as the ‘modifiable area unit problem’ [61,62]. It therefore is likely that there are localized clusters in which public perception of drug problems varies. Finally, the analysis excluded a victimization prognostic as this was not available for this study and should be considered a gap in the analytical framework [36,63].

## 5. Conclusions

Public perceptions of drug problems in London have been trending upwards consistently, rising in 30 out of 32 London boroughs, clustering significantly across North London. It has also been shown that deprivation along with the extent of non-UK-born residents and the level of vehicle crime are all associated with this increase, consistent with the wider literature. Higher rates of burglary are associated with lower levels of perceived local drug problems, suggesting a possible shift in offending behavior. It is argued that people’s perceptions of drug problems within their area are influenced by the social demography of the area and visible manifestations of crime and antisocial behavior.

## Figures and Tables

**Figure 1 ijerph-18-09016-f001:**
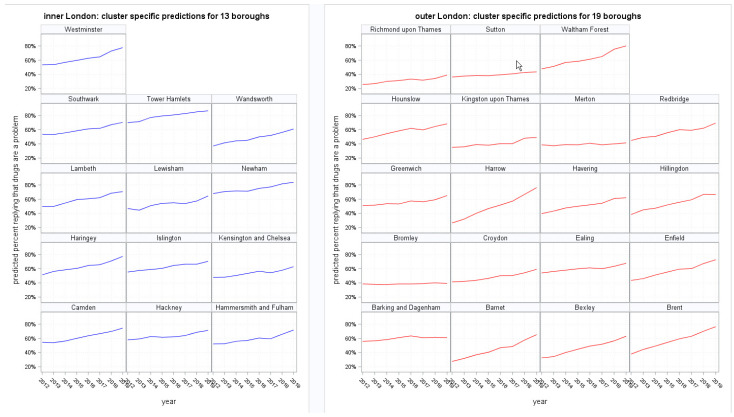
Borough-specific predictions by year and London area: line plots.

**Figure 2 ijerph-18-09016-f002:**
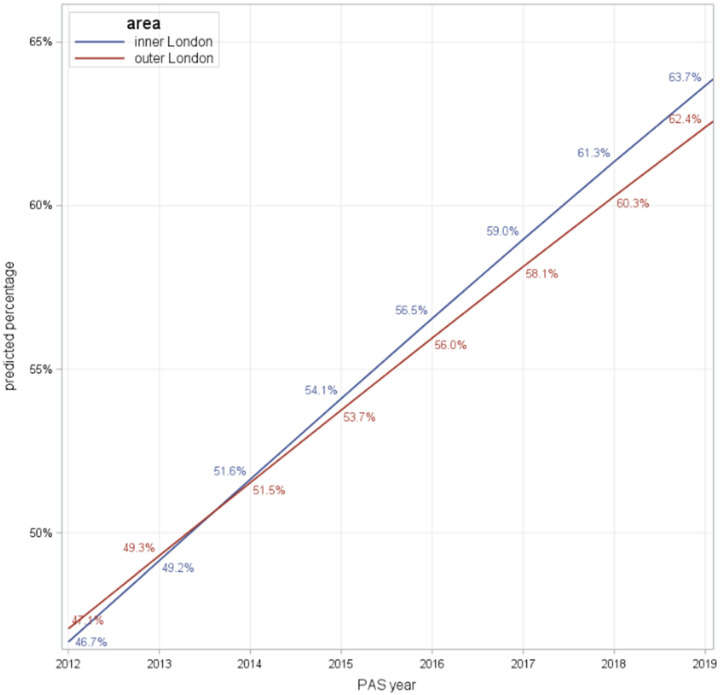
Population-averaged predictions by London area: line plots.

**Table 1 ijerph-18-09016-t001:** Cluster-specific time trend (Odds Ratio sorted in descending order).

Borough	Odds Ratio	OR Lower	OR Upper	T Value	DF	*p*-Value
Harrow	1.3323	1.2279	1.4455	7.18	30	<0.0001
Brent	1.2243	1.1192	1.3392	4.6	30	<0.0001
Waltham Forest	1.2062	1.1042	1.3177	4.33	30	0.0002
Barnet	1.1924	1.096	1.2974	4.26	30	0.0002
Tower Hamlets	1.1814	1.0738	1.2999	3.56	30	0.0012
Bexley	1.1549	1.0647	1.2527	3.62	30	0.0011
Camden	1.1469	1.0525	1.2497	3.26	30	0.0028
Haringey	1.1399	1.0373	1.2526	2.84	30	0.0081
Newham	1.1386	1.0448	1.241	3.08	30	0.0044
Hillingdon	1.135	1.0452	1.2326	3.14	30	0.0038
Enfield	1.1345	1.04	1.2376	2.96	30	0.0059
Wandsworth	1.1281	1.0412	1.2223	3.07	30	0.0045
Islington	1.1145	1.028	1.2082	2.74	30	0.0102
Lambeth	1.1121	1.0157	1.2176	2.39	30	0.0232
Westminster	1.1101	0.9941	1.2397	1.93	30	0.0627
Redbridge	1.1023	1.0201	1.1912	2.57	30	0.0155
Hackney	1.0923	1.0056	1.1865	2.18	30	0.0373
Havering	1.0919	1.006	1.1851	2.19	30	0.0364
Hammersmith and Fulham	1.0914	0.9984	1.193	2.01	30	0.054
Southwark	1.0885	0.9998	1.1851	2.04	30	0.0504
Lewisham	1.0797	0.9941	1.1727	1.9	30	0.0676
Hounslow	1.0741	0.9855	1.1706	1.7	30	0.1001
Croydon	1.0723	0.9774	1.1764	1.54	30	0.1344
Kensington and Chelsea	1.0625	0.9682	1.166	1.33	30	0.1926
Greenwich	1.0547	0.9597	1.1591	1.15	30	0.2587
Ealing	1.0482	0.9648	1.1387	1.16	30	0.2552
Richmond Upon Thames	1.0465	0.9603	1.1405	1.08	30	0.2888
Kingston Upon Thames	1.0428	0.9606	1.132	1.04	30	0.3058
Sutton	1.0352	0.9549	1.1223	0.87	30	0.3885
Barking and Dagenham	1.0102	0.9262	1.1017	0.24	30	0.8135
Merton	0.9849	0.9054	1.0714	−0.37	30	0.7149
Bromley	0.9826	0.9032	1.069	−0.43	30	0.6738

**Table 2 ijerph-18-09016-t002:** Time trend by inner and outer London, Odds Ratios.

Time Trend	Odds Ratio	OR Lower	OR Upper	T Value	DF	*p*-Value
in outer London	1.099	1.026	1.177	2.70	156.70	0.0077
in inner London	1.104	1.020	1.195	2.49	94.34	0.0144

**Table 3 ijerph-18-09016-t003:** Population-averaged fixed effects modeled to the rate of public perceptions of drug use in London, Odds Ratios (OR).

Obs	Prognostic	Estimate	Standard Error	DF	T Value	*p*-Value	Odds Ratio	OR Lower	OR Upper
1	Percentage Non-White	0.0124	0.0045	236	2.75	0.0064	1.0124	1.0035	1.0214
2	Percentage 15–29 years	0.0001	0.0037	236	0.03	0.9784	1.0001	0.9928	1.0075
3	Percentage Non-UK-Born	−0.0055	0.0048	236	−1.15	0.2520	0.9945	0.9851	1.0040
4	EIMD	0.0189	0.0082	78.4	2.31	0.0238	1.0191	1.0026	1.0359
5	Burglary Rate	−0.0392	0.0158	236	−2.48	0.0137	0.9616	0.9321	0.9919
6	Arson/Criminal Damage Rate	0.0129	0.0439	236	0.29	0.7694	1.0130	0.9291	1.1043
7	Drug Offences Rate	−0.0171	0.0157	117	−1.09	0.2788	0.9830	0.9528	1.0142
8	Robbery Rate	0.0127	0.0275	76.9	0.46	0.6457	1.0128	0.9589	1.0697
9	Sexual Offences Rate	0.0599	0.0988	236	0.61	0.5446	1.0618	0.8740	1.2898
10	Theft Rate	0.0486	0.0340	78.3	1.43	0.1570	1.0498	0.9811	1.1233
11	Vehicle Crime Rate	0.0177	0.0086	236	2.07	0.0400	1.0179	1.0008	1.0353
12	Violence Rate	−0.0069	0.0121	236	−0.57	0.5716	0.9932	0.9698	1.0171
13	Weapon Possession Rate	0.0217	0.0180	236	1.21	0.2289	1.0219	0.9864	1.0587
14	Rate in Drug Treatment	0.0610	0.0320	116.8	1.90	0.0594	1.0629	0.9976	1.1324
15	Drug Poisioning Rate	−0.0012	0.0092	236	−0.13	0.8973	0.9988	0.9808	1.0172
16	IP Hospital Admissions Rate	0.0012	0.0028	236	0.44	0.6582	1.0012	0.9957	1.0068

**Table 4 ijerph-18-09016-t004:** Type III tests of fixed effects.

Predictors	Prognostic	Num DF	Den DF	F Value	Pr > F
1	Year	1	156.7	7.28	0.0077
2	Area; inner/outer London	1	41.57	0.01	0.9213
3	Year by area	1	35.29	0.14	0.7114
4	Percentage non-white	1	236	7.58	0.0064
5	Percentage 15–29 year olds	1	236	0.00	0.9784
6	Percentage not born in UK	1	236	1.32	0.2520
7	EIMD	1	78.41	5.32	0.0238
8	Burglary rate	1	236	6.16	0.0137
9	Arson/criminal damage rate	1	236	0.09	0.7694
10	Drug offences rate	1	117	1.18	0.2788
11	Robbery rate	1	76.92	0.21	0.6457
12	Sexual offences rate	1	236	0.37	0.5446
13	Theft rate	1	78.31	2.04	0.1570
14	Vehicle crime rate	1	236	4.27	0.0400
15	Violence rate	1	236	0.32	0.5716
16	Weapon possession rate	1	236	1.45	0.2289
17	Drug treatment rate	1	116.8	3.63	0.0594
18	Drug poisoning rate	1	236	0.02	0.8973
19	Hospital in-patient admissions for drugs rate	1	236	0.20	0.6582

## Data Availability

Data used for this study is available on request.

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
