# Peer review of "A Small Area Estimation Method for Investigating the Relationship between Public Perception of Drug Problems with Neighborhood Prognostics: Trends in London between 2012 and 2019"

_ijerph, 2021, doi:10.3390/ijerph18179016_

Round 1
Reviewer 1 Report
This study examined time-varying factors that are associated with changing perceptions on the drug use problem within the Greater London area from 2012 to 2019.
1) My biggest concern is whether this article is within the scope of IJERPH and the special issue. Please add to the Introduction and Discussion about the relationship between public health and public perception of drug problems.
2) “recorded drug offences in an area were not associated with public perceptions of drug problems” (L.230): “The number of people in specialist drug treatment, the number of drug poisonings and the number of acute in-patient hospital admissions were all not associated with perceived levels of drug issues” (L.237)
I am afraid that public perception represented general fear of crime, not specific to the drug problem. You may be able to obtain the same results even when using public perception of other crime than drugs as the dependent variable.
3) You discussed why two offences were associated with increased perceptions of drug problems, but not why other crimes were not. Why were only those two offences significant and the others non-significant? Without explaining it, your discussion is unconvincing.
4) Would you show the results of other exploratory variables in a table? On the other hand, you can omit Figure 2, which overlaps with table 1.
5) “the percentage of local population who were non-UK born and recorded vehicle crime rates were shown to be positively associated with the public’s perception of drug problems” (L.19): “The percentage of the population that was non-white was shown to be a significant factor” (L.191)
Did you use "non-UK born" and "non-white" interchangeably?
Minor points
1) Sometimes the odds ratio does not match between the main text and tables (maybe because the numbers were rounded off).
2) IMD (L.194) should be EIMD.
Reviewer 2 Report
The paper approaches an important topic - that of investigating factors that correlate with public perceptions in London boroughs . However, I find that the paper, in its present state, has several shortcomings that must be fixed before the paper can be accepted.
-Line 69-71: "Data for.....area." This sentence should be placed in Section 2 as a supplementary explanation of the data
-Line 97-98: "All data ....sources" Specific data sources like the data website need to be written clearly.
-Line 98-100:"The datasets... to 2019" To be more clearly, the explanatory variables can be listed in a table.
-Line 113: What are the 17 prognostics?
-Line 115-119: Why discuss inner London and outer London separately?Are there any differences in their geographical or social environment?
-Line 124-137:The key mathematical expression should be briefly stated, and the input and output variables used in the paper need to be specified
-Line 151-156:The Borough names and ORs can be marked on the map so that the differences between boroughs will be shown more intuitively
-Figure 1: What does the "year 2000 and ..." in the X axis mean?
-Figure 3:It is a non-standard map,lacking legend and north arrow.
-Line 194: What does IMD mean? Or it may be EMID?
Round 2
Reviewer 1 Report
This manuscript has been revised appropriately.
The authors should have indicated where the changes have been made (using the line numbers as a reference) in their responses to the reviewers.
Reviewer 2 Report
The author gives basically satisfactory answers to the previous questions.